# Reducing the Number of Times Eating Out Helps to Decrease Adiposity (Overweight/Obesity) in Children

**DOI:** 10.3390/nu16172899

**Published:** 2024-08-30

**Authors:** Arturo Parra-Solano, Minerva Hernández-Flores, Bernarda Sánchez, Carolina Paredes, Luis Monroy, Florinda Palacios, Laura Almaguer, Ana Rodriguez-Ventura

**Affiliations:** Coordinación de Nutrición y Bioprogramación, Instituto Nacional de Perinatología, Mexico City 11000, Mexico; arturoparrasolano@gmail.com (A.P.-S.); minervahernandez.flores@gmail.com (M.H.-F.); emiberna@yahoo.com.mx (B.S.); caro.90.paredes@gmail.com (C.P.); nutriologobeto@gmail.com (L.M.); sacbenutricion@gmail.com (F.P.); lau.almaguer22@gmail.com (L.A.)

**Keywords:** adiposity, overweight, obesity, children, adolescents, habits, eating out, skipping, sedentary screen time

## Abstract

Adiposity is a chronic disease that must be treated from childhood. Despite the transcendence of improving habits, few interventions report their contribution to decreasing adiposity. Methods: This cohort enrolled children and teens of any gender, 8–18 years old, and with a body mass index (BMI) z-score of ≥1 into “Sacbe”, a comprehensive program to identify which eating habits could reduce BMI z-score. The sample size calculated was 110 participants. We recorded anthropometric measures, clinical history, and habits. A clinically relevant reduction in BMI z-score was defined as ≥0.5 over 12 months or its equivalent according to the months of follow-up. Results: 58.2% were female, the median age was 12 years (range: 9.1–14.7), and the mean BMI z-score was 2.30 ± 0.83. The 82.7% achieved a reduced BMI z-score but 41.8% achieved a clinically relevant reduction with a median follow-up of 6.7 months. Eating out less than once per week was associated with this outcome, even after adjusting for energy intake, other eating patterns, sedentary screen time, physical activity, sleep duration, and sitting time (HR 2.12; 95% CI: 1.07–4.21). Conclusions: Eating out < once/week implicates less processed food exposition and better quality of food; this habit could be the most effective to reduce childhood adiposity.

## 1. Introduction

Adiposity-based chronic disease [1] (overweight or obesity) has increased dramatically in the pediatric population [2]. Multiple unhealthy habits contribute to developing adiposity, principally poor-quality food and the imbalance between the intake and expenditure of energy. Many authors have recommended focusing on improving meal and lifestyle habits to decrease body mass index (BMI) [3], but usually interventions do not report outcomes related to healthy habits. Meal habits make reference to the frequency, spacing, regularity, skipping, schedule, or context (eating out) about food [4].

In Western societies, it is typical to consume three main meals per day; the skipping of a meal would imply eating less and being in a prolonged fasting state [5,6]. The consumption of three meals a day has been associated with greater satiety during the day due to the smaller inter-meal interval. In order to sustain satiety, energy intake should take place during the day, synchronized in an optimal way with energy expenditure. This pattern flattens fluctuations in plasma glucose and insulin concentrations, resulting in a lesser uptake of fatty acids by adipocytes. Moreover, a high meal frequency enables individuals to compensate more accurately for energy deficits or excesses [7]. A higher meal frequency and regular eating pattern seems to be more advantageous than taking meals irregularly and seldom [5].

The evidence supports that obese children are more likely to eat smaller breakfasts than leaner children [3]. Individuals who skip breakfast tend to overeat later in the day, predisposing them to weight gain [3,8]. Prospective studies have found that children and adolescents who use fast food services more frequently experience the greatest increase in BMI z-score [9]. Food consumed out of the home has been associated with poor nutritional quality, high caloric density, and very large portions, promoting an excessive consumption of energy. The evidence indicates that fast food intake is associated with higher weight in children and that weight increases the most when fast food is consumed once or more than once a week [10,11].

Eating speed has an impact on the amount of food eaten and satiety [12]. Eating rapidly enhances the excess intake, reducing the conscious recognition of foods bypassing the internal signals of satiation, resulting in energy disequilibrium and weight gain. On the contrary, eating slowly can promote the secretion of anorexigenic peptides, which enhance satiation and the termination of eating that aids in energy intake regulation [13,14]. The modification of eating quickly could help treat adiposity [15,16]. All these deleterious patterns together have increased in recent decades and are risk factors for the development of adiposity [17,18]; therefore, the cessation of this exposure could decrease adiposity. In order for an individual to acquire a pattern, the conduct has to be repeated over time to account for consistency [19]. The aim of this study was to identify which eating habits could help to reduce BMI z-score.

## 2. Materials and Methods

This is a longitudinal study of a cohort of pediatric participants of “Sacbe”, a comprehensive program that began in 2013 and promotes healthy eating and lifestyle habits to decrease BMI z-scores in children and adolescents. Institutional review board approval (212250-49571) 21 March 2013 and written informed consent from parents and youth were obtained.

Briefly, Sacbe is a comprehensive program with a Pediatric Endocrinologist, nutritionists, psychologists, and a physical activator. The physician explains to groups of families in the first Medical workshop the concept and complications of adiposity and evidence of a healthy lifestyle to prevent diabetes, and in a second Nutritional workshop about healthy food and the size of portions of each group of food. In addition, each nutritionist reviews each family and reinforces the information in the workshops but adapted according to their sociocultural context in each visit. Then, each member of family receives an isocaloric diet designed by registered dietitians. Lifestyle and nutritional counseling was given to parents and their children each month during the first three months and bimonthly afterwards until one year of follow-up [20].

The inclusion criteria were as follows: Female or male gender, 8 to 18 years old, and with a BMI (body mass index) z-score ≥ 1. The exclusion criteria included having any other chronic disease and/or taking any medication related to their weight. Participants were recruited from nearby local schools and the Pediatrics Department at the National Institute of Perinatology.

### 2.1. Assessments

At each clinical visit, a physician recorded clinical history and registered dietitians obtained data regarding meal and lifestyle patterns and a recall of 24 h. They also recorded the height and body weight of each member of the family participating. A BMI z-score was computed with the use of WHO software AnthroPlus 1.0.4, 2007.

The number of eating episodes per day, eating breakfast in the first 2 h of waking up (days per week), eating out (times per week or month), eating speed (average minutes in each meal), physical activity, and sitting time (minutes or hours per day or week) were recorded in each follow-up visit. Energy intake was assessed using 24 h dietary recalls, recording the amount and type of all foods and drinks consumed during the previous day in each visit of follow-up.

### 2.2. Percentage of Adequacy of Energy Intake

We used the Institute of Medicine equation to estimate the total energy expenditure (TEE) in participants (ages 3–18) in a weight maintenance situation. Dietary energy intake above or below the TEE will result in a positive or negative energy balance, respectively. Energy balance is achieved when intake ranges between 90 and 110%.

### 2.3. Sedentary Screen Time

The time spent on sedentary activities in front of a screen (tv/tablet/cellphone/computer) on weekdays and weekends were recorded to obtain the mean sedentary screen time per day.

### 2.4. Sleep Duration

Sleep duration was computed using the averages of participants’ self-reported usual sleep duration on weekdays and weekends.

### 2.5. Mean Difference

To aggregate across all follow-up visits, energy intake, physical activity, adequacy of dietary energy intake, sleep duration, sedentary screen time, and sitting time were averaged by participant all follow-up measurements. The average follow-up (adequacy of dietary energy intake, sedentary screen time, sleep duration, and sitting time) and last observation values (age, weight, height, and BMI z-score) were subtracted from the baseline values to obtain the magnitude of change in these variables.

### 2.6. Clinically Relevant BMI z-Score Reduction

A BMI z-score decrease was obtained by subtracting the last available of the follow-up to the baseline value. Decreases ≥ −0.5 over 12 months or the equivalent in accordance with the months of follow-up was categorized as a clinically relevant BMI z-score reduction [21,22].

### 2.7. Data Analysis

Using the data of the first 170 children and adolescents who were enrolled this program, we analyzed the relationship between acquired healthy meal patterns and clinically relevant BMI z-score reduction. Fourteen participants with missing data and 46 lost to follow-up were excluded from the analysis. The final sample thus consisted of 110 participants. The analysis was carried out using SPSS 22.0 software. The quantitative and categorical variables are reported as appropriate. Multiple imputation was performed when missing values for a given variable did not exceed 10%, excluding independent and dependent variables.

To compare differences, the paired Student’s *t*-test or Wilcoxon rank sum test was used for quantitative variables when appropriate, and McNemar’s test was used for categorical variables.

Differences between groups were derived from the unpaired Student’s *t*-test or Man–Whitney’s U test for quantitative variables when appropriate. The main independent variables were the acquired healthy meal patterns, and the main dependent variable was the achievement of clinically relevant BMI z-score reduction. A set of analyses was performed to determine the association. Relative risk (RR) and 95% confidence intervals (95% CIs) were estimated for each acquired healthy meal pattern individually. Multivariate-adjusted analysis with Cox proportional hazards models was used to derive the hazard ratio (HR) and its 95% CI using three models: Model 1 adjusted for all confounding factors which indicated in a bivariate analysis a significant relationship with the independent and/or dependent variables; Model 2 Model 1 + adjusted simultaneously with every acquired healthy meal pattern; Model 3 Model 2 + adjusted for confounding factors of clinical relevance. Potential confounding factors included the following: change in percentage of adequacy of energy intake, sedentary screen time, sleep duration, and sitting time. Statistical significance was set at an alpha level of 0.05 for all tests.

## 3. Results

In total, 58.2% of participants were female with a median age of 12 years (range: 9.1–14.7), weight 56.5 (43.3, 73.1) kg, height 150 (135.8, 158.4) cm, and a mean 2.3 ± 0.83 BMI z-score, and 38.2% presented overweight (BMI z-score 1–1.99) and 61.8% obesity (BMI z-score 2 or more). The median follow-up time was 6.7 (3.5, 9.8) months, the median of visits was 5 (4–7). The median caloric intake was 1475 (1205, 1877) kcal, with an estimated energy requirement of 2107 (1757, 2491) kcal, considering the level of physical activity, and the percentage of adequacy of energy intake was 71.8 (53.0, 89.2) %. The median time spent physical activity was 17 (5, 43) minutes, along with 7.2 (5.0, 9.0) hours of sitting time, and 3.5 (2.5, 5.0) hours of sedentary screen time, and only 60.7% met the recommended sleep duration per day. The baseline frequency of healthy meal patterns among the total sample was 78.7% for ≥3 eating episodes per day, 54.5% for eating breakfast in the first 2 h after waking up, 47.3% for eating out < once per week, and 70.9% for eating rate ≥ 20 min (Table 1).

Age, weight, height, and physical activity increased, whereas BMI z-score, caloric intake, percentage of adequacy of energy intake, and sitting time decreased. BMI z-score decreased in 82.7% of participants, but its clinically relevant reduction was present in 41.8%. The “healthy” meal patterns increased significantly (*p* < 0.005); three or more eating episodes per day increased 13.8%, having breakfast in the first two hours after waking up increased 24%, and eating speed ≥ 20 mi, 13.5% (Table 1). We compared differences between group with clinically relevant BMI z-score reduction and the group without relevant BMI z-score reduction. A statistically significant difference as observed with weight (−1.1 kg vs. 2.2 kg; *p* < 0.001), BMI z-score (−0.43 DE vs. −0.03 DE; *p* < 0.001), and TEE (−18 kcal vs. 51 kcal; *p* = 0.002) (Table 2).

Children and adolescents who acquired the “healthy” meal pattern of eating out < 1 occasion per week were more likely to attain the clinically relevant BMI z-score reduction, even after adjusting simultaneously with every acquired healthy meal pattern and every confounding factor considered (HR 2.12; CI 95%: 1.07–4.21), as shown in Table 3.

## 4. Discussion

The principal contribution of our research was to analyze the impact of eating habits to reduce BMI z-score in children. Behavior change is the most important factor in chronic disease prevention and treatment. In this study, we found that less times per week eating out was associated with clinically relevant BMI Z-score reduction, and the possible explanation is the reduction in exposition of fast food and ultraprocessed meals [10,11,12]. In fact, four industries (tobacco, unhealthy food, fossil fuels, and alcohol) are responsible for at least a third of global deaths per year [23].

On the other hand, it is really promising that 82.7% of children reduced their PZIMC; we could maintain almost the same level of success comparing our first 55 children studied [21]. In addition, some authors have mentioned that maintaining PZIMC is already a success because the tendency is for it to increase over time [24,25,26,27]. However, according to Reihner [22], who recommends a 0.5 decrease in the PZIMC in one year or proportional to the follow-up time, we identified that only 41.8% achieved it, which is still positive in contrast to the 21% mentioned in the meta-analysis [28,29].

Healthy meal habits increased among the entire sample significantly, greater increases were observed among youth achieving clinically relevant BMI z-score reduction, mainly the pattern of eating breakfast within 2 h after waking up and eating out of the home < once per week. This is also of clinical relevance, since observational studies report that healthy meal habits tend to worsen as we age [30].

Eating out < once per week (or ≤3 times per month) increased the probability of clinically relevant BMI z-score reduction independently of changes in energy balance, sedentary screen time, sitting time, sleep duration, and other “healthy” meal patterns acquired. This result is consistent with other studies that reported that participants who did not eat at fast food restaurants were more successful at weight loss maintenance (OR 1.62; CI 95%: 1.09–2.42) [31,32]. Eating out < once per week (or ≤3 times per month) increased the probability of clinically relevant BMI z-score reduction independently of changes in energy balance, sedentary screen time, sitting time, sleep duration, and other “healthy” meal patterns acquired. This result is consistent with another study that found BMI z-score reductions in those who reported not eating out [30,31]. Fast food consumption has also been associated with poorer metabolic profiles among adolescents and adults [15,24].

The context of relatively more pronounced caloric restriction observed in the group without clinically relevant BMI z-score reduction, together with eating out, could indicate that a state of negative energy balance coupled with episodes of hypercaloric intake promotes fat storage. Ochiai and colleagues evaluated the pattern of eating speed and found that anthropometric variables were significantly worse among those who continued an unhealthy meal pattern [33], but we did not find an association with this habit. The longitudinal design and the analysis we performed ensured the temporal progression of exposure and outcome. Independent variables were modeled to reflect the effect of short-term stable “healthy” meal patterns.

The areas of opportunity in which to improve our study are to analyze more accurately the quality of diet and enlarge the sample size to consider other variables like age subgroups, categories of BMI, sex, and other sociodemographic variables.

In conclusion, this family approach and interdisciplinary study inside the comprehensive program “Sacbe” remained effective in decreasing BMI z-score value, and in a very high percentage of participants. Almost every habit improved, but eating out less than once per week could be the best strategy to reduce BMI z-score by decreasing ultraprocessed meal exposition and improving the quality of food prepared at home.

## Figures and Tables

**Table 1 nutrients-16-02899-t001:** Comparison of anthropometric characteristics and lifestyle habits (baseline to follow-up).

Characteristics and Habits	Baseline Visit Median (IQR)	Follow Up Median (IQR)	*p*
Age (years)	12 (9.1, 14.7)	12.7 (9.6, 15.2)	<0.001 ^†^
Weight (kg)	56.5 (43.3, 73.1)	57.8 (44.7, 73.6)	0.006 ^†^
Height (cm)	150.0 (135.8, 158.4)	152.4 (139.7, 161.1)	<0.001 ^†^
BMI z-score (SD)	2.30 ± 0.83	2.10 ± 0.79	<0.001 *
Energy intake (kcal/day)	1475 (1205, 1877)	1331 (1124, 1565)	0.001 ^†^
Energy estimated requirement	2107 (1757, 2491)	2148 (1768, 2669)	0.093 ^†^
Physical activity (min/day)	17 (5, 43)	26 (14, 44)	0.011 ^†^
Percentage of adequacy of energy intake (%)	71.8 (53.0, 89.2)	62.9 (51.0, 75.2)	<0.001 ^†^
Sleep duration (h/day)	9.0 (7.9, 9.5)	8.9 (7.9, 9.4)	0.936 ^†^
Sedentary screen time (h/day)	3.5 (2.5, 5.0)	3.3 (2.4, 4.8)	0.302 ^†^
Sitting time (h/day)	7.2 (5.0, 9.0)	6.7 (5.0, 8.0)	0.008 ^†^
	N (%)	N(%)	*p*
Obesity	68 (61.8%)	61 (58.1%)	0.092 ^‡^
Recommended sleep	65 (60.7%)	58 (54.2%)	0.360 ^‡^
≥3 eating episodes/day	85 (78.7%)	99 (92.5%)	0.001 ^‡^
Eating breakfast waking up	60 (54.5%)	84 (78.5%)	<0.001 ^‡^
Eating out < once/week	52 (47.3%)	65 (59.6%)	0.029 ^‡^
Eating speed ≥ 20 min	78 (70.9%)	92 (84.4%)	0.014 ^‡^

^†^ Wilcoxon rank sum test, * paired Student’s *t*-test, and ^‡^ McNemar.

**Table 2 nutrients-16-02899-t002:** Comparison of anthropometric and lifestyle characteristics among groups with presence or absence of the outcome (baseline to follow-up within groups, delta’s between groups).

Characteristics	Clinically Relevant BMI z-Score Reduction (n = 45)			Not Enought BMI z-Score Reduction (n = 65)			*p*
	Baseline	Follow-Up	Difference	Baseline	Follow-Up	Difference	
Age (years)	11.9 (8.8, 15.2)	12.4 (9.2, 15.4) *	0.3 (0.2, 0.7)	12.0 (9.5, 14.5)	12.8(10.2, 15.2) *	0.7 (0.4, 0.9)	<0.001 ^†^
Weight (kg)	53.3 (42.4, 74)	53.5 (42, 72) *	−1.1 (−2.7, −0.1)	56.9 (43.5, 72.6)	59.5(47.4, 74.6) *	2.3 (0.8, 4.1)	<0.001 ^†^
Height (cm)	150 (133, 158)	152 (136, 160) *	1.5 (0.4, 3.0)	150 (138, 160)	153 (141, 161) *	2.5 (0.9, 3.4)	0.082 ^†^
BMI z-score (SD)	2.48 ± 1.1	2.07 ± 1.0 **	−0.43 ±0.29	2.16 ± 0.58	2.13 ± 0.56 **	−0.05 ± 0.17	<0.001 ^‡^
Energy intake (kcal/d)	1461 (1126, 1948)	1264 (1103, 1509) *	−201 (−388, 86)	1475 (1245, 1846)	1397 (1189, 1661) *	−213 (−477, 237)	0.901 ^†^
Energy requirement	2152 (1779, 2545)	2126 (1770, 2521)	−23 (−128, 67)	2066 (1737, 2476)	2215 (1742, 2755) *	52 (−24, 342)	0.001 ^†^
Physical activity (min/day)	17 (0, 40)	23 (12, 38)	4 (−13, 17)	17 (9, 43)	29 (14, 46) *	5 (−7, 22)	0.632 ^†^
% of Adequacy energy intake	68.2 (54.2, 85.8)	63.7 (50.4, 74.2) *	−3.7 (−25.3, 5.4)	73.9 (52.3, 91.6)	62.6 (50.9, 77.3) *	−11.8 (−28.7, 7.1)	0.541 ^†^
Sleep duration (h/day)	9.0 h (7.8, 9.9)	9.0 h (8.2, 9.5)	−6 min (−57, 53)	8.9 h (7.9, 9.3)	8.8 h (7.6, 9.4)	9 min (−38, 38)	0.806 ^†^
Screen time (h/day)	3.5 h (2.9, 5.1)	3.3 h (2.2, 5.0)	−30 min (−110, 42)	3.0 h (2.0, 4.4)	3.3 h (2.5, 4.7)	−1 min (−88, 90)	0.161 ^†^
Sitting time (h/day)	7.2 h (5.0, 9.0)	6.4 h (5.0, 7.5) *	−35 min (−163, 58)	7.0 h (5.0, 9.0)	7.1 h (5.5, 8.0)	−35 min (−149, 55)	0.682 ^†^
Follow-up (months)		4.1 (2.1, 7.6)			7.9 (5.2, 10.8)		<0.001 ^†^

* *p* < 0.05 Wilcoxon rank sum test, ** *p* < 0.05 paired Student’s *t*-test. ^†^ Mann–Whitney’s U test, and ^‡^ unpaired Student’s *t*-test.

**Table 3 nutrients-16-02899-t003:** Association between acquiring healthy meal patterns and clinically relevant BMI z-score reduction.

		HR (CI 95%)		
	RR (CI 95%) Unadjusted	Model 1	Model 2	Model 3
Eating ≥ 3 meals per day	1.51 (0.95–2.41)	NA	1.08 (0.54–2.18)	1.13 (0.56–2.28)
Eating breakfast in the first 2 h of waking up	1.25 (0.80–1.95)	1.11 (0.6 -2.07) ^‡^	1.17 (0.62–2.21)	1.14 (0.60–2.17)
Eating out < once/week	1.61 (1.03–2.50)	NA	2.01 (1.03–3.95)	2.12 (1.07–4.21)
Eating ≥ 20 min	0.77 (0.41–1.42)	0.96 (0.43-2.16) ^†^	0.88 (0.37–2.09)	0.90 (0.38–2.13)

Model 1: Acquired healthy meal patterns separately and adjusted for statistically confounding factors (NA: not applicable), ^‡^ change in adequacy of dietary energy intake (%), and ^†^ sedentary screen time (min/day). Model 2: Acquired healthy meal patterns included simultaneously. Model 3: Model 2 + adjusted for clinically confounding factors (change in sleep duration (min/day) and sitting time (min/day)).

## Data Availability

The data are unavailable due to privacy.

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
