# Peer review of "Reducing the Number of Times Eating Out Helps to Decrease Adiposity (Overweight/Obesity) in Children"

_nutrients, 2024, doi:10.3390/nu16172899_

Round 1

Reviewer 1 Report

Comments and Suggestions for Authors

Please take note of the following summary: 
1. The authors state that 82.7% of participants saw a decrease in BMI z-score, and only 41.8% experienced a clinically significant reduction. This raises concerns about the significance of the observed changes and whether they align with the intervention's goals. 
2. The authors discuss statistically significant differences between groups for various parameters but do not address potential confounding factors or limitations of these statistical comparisons. A more comprehensive discussion of these limitations could enhance the interpretation of the results. 
3. Associations between specific healthy meal patterns and clinically relevant BMI z-score reduction are mentioned, but potential confounders and other influencing variables are not considered.
4. The manuscript lacks detailed information about specific interventions and strategies used in the "Sacbe" program, which could hinder assessing its effectiveness and potential for replication in other settings.
5. The 12-month follow-up period may be relatively short for a chronic condition like adiposity. It would be valuable to assess the longer-term effects of the intervention to determine its sustainability and impact over time.

Author Response

Thank you very much for taking the time to review this manuscript. Please find the detailed responses below and the corresponding revisions/corrections highlighted/in track changes in the re-submitted files. We agree with your great recommendations. Research design is good but definitively we need more time of follow-up, our result will be best explained and also our conclusions.

Point-by-point response to Comments and Suggestions for Authors

Comments 1: The authors state that 82.7% of participants saw a decrease in BMI z-score, and only 41.8% experienced a clinically significant reduction. This raises concerns about the significance of the observed changes and whether they align with the intervention's goals. 

Response 1:  We consider that the fact that 82.7% of children reduced their PZIMC is very promising because some authors and meta-analyses report success in only 21% of children and it has been mentioned that even maintaining PZIMC is already a success because the tendency is for it to increase over time. However, according to Reihner, who recommends a 0.5 decrease in the PZIMC in one year or proportional to the follow-up time, we identified that only 41.8% achieved it, which is still positive in contrast to the 21% mentioned. Thank you for pointing this out. We agree with this comment. Therefore, we have added this explanation in Discussion, second paragraph.

Comments 2:  The authors discuss statistically significant differences between groups for various parameters but do not address potential confounding factors or limitations of these statistical comparisons. A more comprehensive discussion of these limitations could enhance the interpretation of the results. 

Response 2: The greatest limitation we had when conducting the statistical analyses was undoubtedly our sample size of 110 participants, however it was calculated considering the waited results and the variables of habits related to eat. With our small sample, we aimed to take into account most of the potentially confounding factos to make our “best multivariate adjusted model” as parsimonious as possible, so we had to exclude variables that would otherwise should have been included.

Comments 3:

Associations between specific healthy meal patterns and clinically relevant BMI z-score reduction are mentioned, but potential confounders and other influencing variables are not considered.

Response 3: We conducted an analysis without considering confounding variables in the unadjusted Relative Risk (RR), which is presented in the second column of Table 3. But starting from the third column of table 3 described as Model 1, we carried out a multivariate analysis adjusting for variables that statistically demonstrated to be confounders. In Model 2, we additionally adjusted Model 1 with all the healthy meal patterns acquired during the follow up because they could also explain a change in the BMI z-score. Finally, in the fourth column of Table 3 described as Model 3, we added to Model 2 other confounding factors that the literature has considered as peripheral maneuvers that could also explain the decrease in body mass index

Comments 4:

The manuscript lacks detailed information about specific interventions and strategies used in the "Sacbe" program, which could hinder assessing its effectiveness and potential for replication in other settings.

Response 4: Agree, but we mentioned our paper published in 2018 about the strategies, however, we decided to include more information about it in methods. We have added more information in Materials and Methods, second paragraph.

Comments 5:

The 12-month follow-up period may be relatively short for a chronic condition like adiposity. It would be valuable to assess the longer-term effects of the intervention to determine its sustainability and impact over time

Response 5: Agree, we have followed these patients and new participants but after the pandemic it has not be easy to maintain the adherence. Anyway, we continue the followup. Thank you so much for this observation. We hope in the future to get more patients and budget.

Reviewer 2 Report

Comments and Suggestions for Authors

I would like to thank the editor for the opportunity to read this work.

Abstract

The first sentence in the methods section is unclear.

Introduction

The introduction is well described, however it is not so clear the study aim. Particularly, based on Manuscript title the goal is different. Moreover, I suggest to add the innovative aspect of the work and the hypothesis.

Methods

Well described however I do not understand if in this study some intervention was applied. In this case this can be a quasi-experimental study with a single experimental group even considering the first analysis in which authors compared baseline with follow-yup.

Results

How did authors calculate the obesity? Did they use the BMI with Cole et al. categories?

Discussion

The first line of the discussion is not congruent with the aim of the study wrote in the introduction.

Discussion should be revised, now it is quite a repetition of the results section.

Comments on the Quality of English Language

Minor editing of English language required

Author Response

Thank you very much for taking the time to review this manuscript. Please find the detailed responses below and the corresponding revisions/corrections highlighted/in track changes in the re-submitted files. We agree with your great recommendations.

Abstract. The first sentence in the methods section is unclear.

We already modified this first sentence in the methods on the abstract (14 and 15 lines)

Introduction. The introduction is well described; however, it is not so clear the study aim. Particularly, based on Manuscript title the goal is different. Moreover, I suggest to add the innovative aspect of the work and the hypothesis.

That’s right, thank you so much, we already added this innovative aspect and the hypothesis (lines 64 and 65).

Methods. Well described however I do not understand if in this study some intervention was applied. In this case this can be a quasi-experimental study with a single experimental group even considering the first analysis in which authors compared baseline with follow-up.

The intervention was already described in our past article in 2018, Sacbe is a comprehensive program, 2 workshops are explained in small groups of families by the pediatric endocrinologist followed of a reinforcement by nutritionists in each visit. In the first 55 participants we effectively considered a quasi-experimental study to find the effectivity of our program but in this second phase, we added 55 more participants and did a longer follow-up, so then, we considered a cohort study because our aim changed, we wanted to identify the principal eating habits related to decrease BMI z-score.

Results. How did authors calculate the obesity? Did they use the BMI with Cole et al. categories?

We defined Overweight if BMI z-score was 1-1.99 and Obesity if it was 2 or more (line 151).

Discussion. The first line of the discussion is not congruent with the aim of the study wrote in the introduction. Discussion should be revised, now it is quite a repetition of the results section.

We already changed the first line of the discussion in order to be congruent (lines 193 and 194). We improved our discussion and added other references and tried do not repeat of results section (lines 200-206).
